# A Warning against the Negligent Use of Cannabidiol in Professional and Amateur Athletes

**DOI:** 10.3390/sports7120251

**Published:** 2019-12-14

**Authors:** Dirk W. Lachenmeier, Patrick Diel

**Affiliations:** 1Chemisches und Veterinäruntersuchungsamt (CVUA) Karlsruhe, Weissenburger Strasse 3, 76187 Karlsruhe, Germany; 2Institute for Cardiovascular Research and Sports Medicine, Department of Molecular and Cellular Sports Medicine, German Sports University, 50333 Cologne, Germany; Diel@dshs-koeln.de

**Keywords:** cannabis sativa, cannabidiol, delta-9-tetrahydrocannabinol, exercise nutritional science, doping in sports

## Abstract

Cannabidiol (CBD) is a non-psychoactive cannabinoid, widely marketed to athletes for claimed effects such as decreased anxiety, fear memory extinction, anti-inflammatory properties, relief of pain and for post-exercise recovery. The World Anti-Doping Agency (WADA) has excluded CBD from its list of prohibited substances. Nevertheless, caution is currently advised for athletes intending to use the compound—except CBD, all other cannabinoids are still on the prohibited list. CBD products, specifically non-medicinal, so-called full-spectrum cannabis extracts, may contain significant levels of these substances, but also contaminations of tetrahydrocannabinol (THC) (>2.5 mg/day in >30% of products on the German market) potentially leading to positive doping tests. Labelled claims about CBD content and absence of THC are often false and misleading. Contaminations with the psychoactive THC can result in adverse effects on cognition and, in general, the safety profile of CBD with respect to its toxicity is a controversial topic of discussion. For these reasons, we would currently advise against the use of over-the-counter CBD products, especially those from dubious internet sources without quality control.

## 1. Introduction

While being illegal in many jurisdictions, cannabis smoking has been claimed as being helpful for some activities such as extreme sports, as it may improve muscle relaxation, reduces anxiety, and help users overcome fear memories (e.g., negative experiences), potentially leading to enhanced performance [1]. For these reasons cannabis has been included on the World Anti-Doping Agency (WADA) list as a substance prohibited in-competition. In fact, there are only a few data regarding its effectiveness and previous reviews concluded that there is a lack of evidence regarding performance-enhancing effects [2,3]. Based on this lack of data and its negative effects on cognition, including impairment of decision making and alertness [1], as well as its status as illegal drug (where applicable), cannabis smoking is clearly not advisable for professional and amateur athletes alike. However, beside all these regulations and concerns regarding its effectiveness, the smoking of cannabis and consumption of supplements containing cannabinoids appears to be popular in public and elite sports.

The potential advantages of cannabis in sports have certainly stimulated interest surrounding cannabidiol (CBD), which is one of the cannabinoid compounds naturally found in *Cannabis sativa*. CBD is structurally related to the main psychoactive compound Δ^9^-tetrahydrocannabinol (THC), which is responsible for the adverse effects of cannabis. CBD is non-psychoactive, but some pre-clinical evidence points to the fact that it may be responsible for some of the advantageous effects of cannabis for sports activities, such as decreased anxiety and fear memory extinction [1]. The further advertised effects of CBD for athletes include anti-inflammatory properties, the relief of arthritis and pain-related behaviors, as well as post-exercise recovery [4]. Consumers’ interest in CBD has considerably increased since 2015 and is still accelerating [5].

According to the World Health Organization (WHO), CBD is an effective treatment of epilepsy, and there may be some preliminary evidence for its application in a number of other medical conditions. The WHO also remarked upon the unsanctioned use of CBD-based products available online for the treatment of many ailments [6]. While CBD is generally well tolerated with a good safety profile [6,7,8], THC-like subjective or physiological effects have been reported. However, these may be due to THC-contamination of CBD-products rather than due to the CBD itself [9]. Nonetheless, some recent animal experiments have raised concerns regarding the hepatotoxicity [10] and reproductive toxicity [11] (see review in [12]).

Cannabidiol is currently approved in the European Union (EU) in a single medicinal product, namely Epidiolex^®^ for the treatment of seizures in patients with two rare, severe forms of childhood-onset epilepsy. However, most of the CBD products worldwide are available as food supplements or compounded foods with CBD or hemp extract as an ingredient [13]. Such unapproved products, which normally do not comply with quality standards, are currently easily available to athletes. A survey of cannabis use patterns in the US detected that older athletes who are newer to cannabis tend to only use CBD. However, in these athletes it provided the least reported benefit (albeit, with the least adverse effects) [14].

The WADA has excluded CBD from its list of prohibited cannabinoids [15]. However, it is important to mention that, according to the 2020 prohibited list of the WADA [15], cannabinoids in general still are listed in group S8, as substances forbidden to be used in competition. Here it is defined that all naturally occurring and synthetic cannabinoids—except CBD—are forbidden. Because most CBD products are sold as so-called full-spectrum products, they also contain other cannabinoids. Therefore, in competition, the use of full-spectrum CBD products is definitely prohibited by WADA. Moreover, the WADA remarked that “athletes should be aware that some CBD products extracted from cannabis plants may also contain THC that could result in a positive test for a prohibited cannabinoid” [15]. The threshold for a positive THC test has been set by WADA to 150 ng/mL of 11-nor-9-carboxy-THC in urine [16]. Other organizations, such as Drug Free Sports International, similarly warned that contaminated CBD products may cause positive urine tests [17].

The possibility of positive drug testing following CBD oil use is certainly more than a hypothetical risk. Studies worldwide have detected residual levels of THC in CBD products [9,18,19,20,21]. Most recently our group has detected THC in dose levels of >2.5 mg/day in 10 out of 28 commercial CBD products from the German internet and retail market [9]. Positive urine drug testing for several days may be expected from daily oral doses of more than 1 mg [13,22]. Therefore, more than 1/3 of CBD products on the German market—if consumed by athletes—would probably lead to false-positive urine doping tests.

This commentary therefore wants to provide a warning against the negligent use of CBD in athletes and provide the following considerations for consumers.

## 2. Considerations for the Use of CBD Products in Athletes

While the authors currently would not endorse the use of CBD in the current situation of the almost uncontrolled and unregulated market, the following points may be considered by athletes still intending to use CBD products
Medicinal products—if available—are to be preferred over food supplements or other product categories such as cosmetics or air fresheners, because medicinal products are typically better purified and more strictly controlled. The use of medicinal products on prescription by sport medicine specialists also has the advantage of medical surveillance for adverse effects such as liver toxicity [12].For products marketed as food or nutritional supplements, the following special scrutiny is required:Products labelled as “full spectrum extracts” contain—besides CBD—undefined concentrations of other cannabinoids. Therefore, according to the regulations of the WADA (prohibited list) [15] the use of such products is, in general, illegal. Athletes should neither use such products in competition nor during competition periods (due to the long window of detectability of 11-nor-9-carboxy-THC in urine).Producers should provide credible analytical proof for the claims about CBD content on the labels. Many studies of CBD products have detected considerable mislabeling of content [9,18,19,20,21]. Analytical reports on websites of CBD producers should therefore be critically questioned, specifically if they only provide “lower than” results or even THC-free claims but with comparably high limits of detection (such as only <0.2%). Limits of detection in the percentage range are not able to ensure the absence of THC in a magnitude that would also exclude THC-positive urine tests. For example, the German target value to exclude any risk of THC in foods would be 150 µg/kg (0.000015%) [22,23,24], and only such a level would also exclude any risk of a positive drug test with certainty. Sensitive methods, such as combinations of gas or liquid chromatography with tandem mass spectrometry (MS/MS), are necessary for adequate quality control [13,25]. Methods with unspecific and insensitive detectors, such as flame-ionization detection (FID) are typically inadequate. Therefore, a careful interpretation of analytical reports is necessary, which is probably impossible for CBD consumers, who should consult experts if in doubt. To avoid the adverse effects of THC, the residual THC content in CBD products must be below the acute reference dose of the European Food Safety Authority (EFSA), which is 1 µg/kg bodyweight/day [26]. The highest risk for residual THC comes from products not intended for human consumption, such as air fresheners or cosmetics, which may be used orally off-label.Good manufacturing practice would also include testing for residual solvents, contaminants and residues such as heavy metals, mycotoxins or pesticides.Mandatory labelling should be complete and correct (e.g., regarding ingredients, minimum durability, etc.).Claims about health- and disease-related effects are prohibited except in cases of approved medicinal products. If producers advertise effects such as cancer-prevention for food supplements, this seriously questions the trustworthiness of the producers, and means that they probably mislead the consumer about other things, such as absence of THC content as well.The safety of the long-term use of CBD is understudied and there are indications of some adverse effects including liver toxicity and an influence of male fertility [12]. A risk-benefit analysis should carefully consider the possible, so far non-approved benefits of CBD for athletes [14] against its safety-profile.

## 3. Conclusions

While CBD itself is not banned (according to WADA), the content of commercial products often contains banned substances such as THC and other cannabinoids, a problem which should present risks that far outweigh any benefits for athletes and their support staff. Therefore, the most appropriate risk-minimization strategy is to avoid all CBD products to prevent, as best as possible, inadvertent doping from these often-contaminated products. From a regulatory point of view, we would strongly advise the introduction of stricter regulations for non-medicinal CBD products, e.g., labelling requirements and mandatory testing (see [21]).

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
