# Peer review of "A Warning against the Negligent Use of Cannabidiol in Professional and Amateur Athletes"

_sports, 2019, doi:10.3390/sports7120251_

Round 1

Reviewer 1 Report

It is a very interesting article. Cannabis (marijuana) is undergoing extensive regulatory review in many global jurisdictions for medical and nonmedical access. Cannabis has potential impact on the health of athletes as well as on performance in both training and in competition. Although cannabis use is more prevalent in some athletes engaged in high-risk sports, there is no direct evidence of performance-enhancing effects in athletes.

Many cannabis and CBD products are available over the internet. However, the Food and Drug Administration (FDA) have approved none of these artisanal preparations (irrespective of THC content) for safety or efficacy, and state laws requiring quality control are inconsistent and sometimes nonexistent.

And it`s very important the WADA remarked that “athletes should be aware that some CBD products extracted from cannabis plants may also contain THC that could result in a positive test for a prohibited cannabinoid”.

In addition, it would be important to highlight the possible negative effects on the athlete's health.

Kennedy MC1. J Sci Med Sport. 2017 Sep;20(9):825-829. Cannabis: Exercise performance and sport. A systematic review.

Ware MA, Jensen D, Barrette A, Vernec A, Derman W. Clin J Sport Med. 2018 Sep;28(5):480-484. Cannabis and the Health and Performance of the Elite Athlete.

Author Response

It is a very interesting article. Cannabis (marijuana) is undergoing extensive regulatory review in many global jurisdictions for medical and nonmedical access. Cannabis has potential impact on the health of athletes as well as on performance in both training and in competition. Although cannabis use is more prevalent in some athletes engaged in high-risk sports, there is no direct evidence of performance-enhancing effects in athletes.

Many cannabis and CBD products are available over the internet. However, the Food and Drug Administration (FDA) have approved none of these artisanal preparations (irrespective of THC content) for safety or efficacy, and state laws requiring quality control are inconsistent and sometimes non-existent.

And it`s very important the WADA remarked that “athletes should be aware that some CBD products extracted from cannabis plants may also contain THC that could result in a positive test for a prohibited cannabinoid”.

Response: Thank you for your positive remarks.

In addition, it would be important to highlight the possible negative effects on the athlete's health.

Response: We have added information on the lack of evidence of effects and included the references below. Thank you for pointing these out. The detrimental effects are pointed out in line 33.

Kennedy MC1. J Sci Med Sport. 2017 Sep;20(9):825-829. Cannabis: Exercise performance and sport. A systematic review.

Ware MA, Jensen D, Barrette A, Vernec A, Derman W. Clin J Sport Med. 2018 Sep;28(5):480-484. Cannabis and the Health and Performance of the Elite Athlete.

Reviewer 2 Report

The commentary "A warning against the negligent use of cannabidiol (CBD) in professional and amateur athletes" fits the aim of the journal. However, in order to be publishable the manuscript should be corrected. The "Advice for use of CBD products in athletes" section presents 7 advices which often overlap (for example advices 2 - 7 are consequence of advice 1). Thus, a medicinal product has quality control reports and meets standards of EMA/FDA regulations. On the other hand, supplements and other products are low regulated (and depends on each state laws) and often their quality control report is poor and misleading. Moreover, the market is full of counterfeited products of low quality and with many label issues. I suggest the authors to revise all 7 advices and to study the classification of products with CBD available on market. Conclusions should also be changed accordingly.

Author Response

The commentary "A warning against the negligent use of cannabidiol (CBD) in professional and amateur athletes" fits the aim of the journal. However, in order to be publishable the manuscript should be corrected. The "Advice for use of CBD products in athletes" section presents 7 advices which often overlap (for example advices 2 - 7 are consequence of advice 1). Thus, a medicinal product has quality control reports and meets standards of EMA/FDA regulations. On the other hand, supplements and other products are low regulated (and depends on each state laws) and often their quality control report is poor and misleading. Moreover, the market is full of counterfeited products of low quality and with many label issues. I suggest the authors to revise all 7 advices and to study the classification of products with CBD available on market. Conclusions should also be changed accordingly.

Response: Thank you for the advice. We have renumbered the section, to first discuss medicinal products (#1), then food supplements (#2 with new sub-points i-iv) and finally some general remarks (#3, and #4 with regards to both product categories, i.e. medicinal and food products). We have added the remarks regarding counterfeited products to the conclusion. Thank you for pointing this out.

Reviewer 3 Report

Thank you for the opportunity to review this paper for Sports. While the article is generally well written, I am unsure of the main purpose of the submission as it seems to be biased towards the view that cannabidiol (CBD) should be taken and safeguards put in place to minimise adverse drug tests (e.g., L83 opening with the fact that medicinal products should be sought). First and foremost, cannabis is an illegal drug (i.e., class B in the UK, and alluded to in the manuscript in USA presumably?); this major fact presently features as a secondary critique of the substance in L32, following that of cognitive impairment, when referring to it’s WADA status. Moreover, the article purports benefits of smoking cannabis in the opening line. Notwithstanding again the illegality of its use, minimal references are supported for the benefits when the authors are presumably trying to rationalise the consumption of CBD. While L36 mentions CBD being a subclass of cannabinoids, it mentions minimally the key fact that cannabinoids are prohibited by WADA. Similarly, CBD, as highlighted by the authors, poses more than a hypothetical risk of contamination with banned substances such as THC. Therefore, it is my opinion that it’s consumption is not to be encouraged in any form given the obvious risks of contamination, so a document with a section about advice for use is almost indirectly sanctioning such use within certain parameters. I therefore reject the submission.

Author Response

Thank you for the opportunity to review this paper for Sports. While the article is generally well written, I am unsure of the main purpose of the submission as it seems to be biased towards the view that cannabidiol (CBD) should be taken and safeguards put in place to minimise adverse drug tests (e.g., L83 opening with the fact that medicinal products should be sought).

Response: The authors clearly do not endorse CBD products, this is a misunderstanding. We clarified the text throughout and added a statement to line 83. Nevertheless, it is a fact that athletes are taking CBD and we want to warn against this practice. This was the main intention of our article.

First and foremost, cannabis is an illegal drug (i.e., class B in the UK, and alluded to in the manuscript in USA presumably?).

Response: Yes, but it depends on jurisdiction, and sometimes even on state law. So we excluded a report on the complicated legal situation of cannabis worldwide. CBD is even more difficult. Notwithstanding legality, both products (i.e. cannabis as marihuana, as well as CBD) are widely available to athletes everywhere.

This major fact presently features as a secondary critique of the substance in L32, following that of cognitive impairment, when referring to it’s WADA status.

Response: The section around line 32 was rewritten and hopefully clarified.

Moreover, the article purports benefits of smoking cannabis in the opening line. Notwithstanding again the illegality of its use, minimal references are supported for the benefits when the authors are presumably trying to rationalise the consumption of CBD.

Response: The sentences about benefits were considerably toned down as requested.

While L36 mentions CBD being a subclass of cannabinoids, it mentions minimally the key fact that cannabinoids are prohibited by WADA. Similarly, CBD, as highlighted by the authors, poses more than a hypothetical risk of contamination with banned substances such as THC. Therefore, it is my opinion that it’s consumption is not to be encouraged in any form given the obvious risks of contamination, so a document with a section about advice for use is almost indirectly sanctioning such use within certain parameters. I therefore reject the submission.

Response: We basically agree with these points (apart from the rejection), as this is exactly the conclusion of our article. CBD may have some benefits. CBD has been taken recently  from the WADA list and therefore is of course of interest to athletes.  In the respective paragraph of the antidoping code it is stated that all other cannabinoids “EXCEPT” CBD are forbidden.  Taking a substance from the forbidden list makes it even more interesting for the athletes. A good example is caffeine which also has been taken from the list, despite having clear performance enhancing effects.  So worldwide more and more athletes are interested in taking CBD, especially in countries or regions where cannabis smoking is not illegal. These athletes must be aware about the risks of taking CBD containing products. This is the main purpose of this article. Contaminated products pose a clear risk and should not be encouraged, see our conclusion. We hope that the clarifications throughout now better show our intentions.

Round 2

Reviewer 2 Report

The authors have made the requested changes and the article can be published.

Author Response

The authors have made the requested changes and the article can be published.

Response: Thank you!

Reviewer 3 Report

As alluded to in the initial review of the manuscript, I have significant concerns over the content of this manuscript in relation to appearing to "advocate" taking CBD - albeit with caution. While CBD itself is not banned (according to WADA), the content of such products often contain banned substances - a fact alluded to in the manuscript, and one which should present risks that far outweigh any benefits for athletes and their support staff. Therefore, the most appropriate risk-minimisation strategy is to avoid all CBD products to prevent, as best as possible, inadvertent doping from these often contaminated products. While I understand that such products are taken, I don't feel it wise for the journal to publish an article with recommendations about taking CBD (albeit well-intentioned) as such evidence could be seen to be advocating consumption of a likely contaminated product. 

Author Response

(x) English language and style are fine/minor spell check required

Response: The text was carefully spell-checked and some typos were corrected. Thank you for the advice.

As alluded to in the initial review of the manuscript, I have significant concerns over the content of this manuscript in relation to appearing to "advocate" taking CBD - albeit with caution. While CBD itself is not banned (according to WADA), the content of such products often contain banned substances - a fact alluded to in the manuscript, and one which should present risks that far outweigh any benefits for athletes and their support staff. Therefore, the most appropriate risk-minimisation strategy is to avoid all CBD products to prevent, as best as possible, inadvertent doping from these often contaminated products. While I understand that such products are taken, I don't feel it wise for the journal to publish an article with recommendations about taking CBD (albeit well-intentioned) as such evidence could be seen to be advocating consumption of a likely contaminated product.

Response: In re-iteration to the response in review round #1, the authors clearly do not endorse CBD products. We also do not want to “advocate” CBD, but rather warn against its risks. However, the reality is that CBD is widely used (according to a new survey, 16% of Europeans reported having used CBD or CBD products [1]). We do not believe that our rather critical commentary can be interpreted by anyone as recommendation about taking CBD. Nevertheless, to accommodate the reviewer, we have looked through the paper and toned down any aspects that might be mis-interpreted as “endorsement” or “advocating” of CBD. Specially, we changed the title of section 2 from “advice” to “consideration”. Perhaps the word “advice” has been open to misinterpretation as “endorsement”. Furthermore, the conclusion was changed to include the reviewer’s suggestion above. Thank you for this suggestion!

[1] https://newfrontierdata.com/marijuana-insights/cbd-across-the-pond-cannabidiol-use-in-europe/